# Associations between Changes in Health Behaviours and Body Weight during the COVID-19 Quarantine in Lithuania: The Lithuanian COVIDiet Study

**DOI:** 10.3390/nu12103119

**Published:** 2020-10-13

**Authors:** Vilma Kriaucioniene, Lina Bagdonaviciene, Celia Rodríguez-Pérez, Janina Petkeviciene

**Affiliations:** 1Faculty of Public Health, Medical Academy, Lithuanian University of Health Sciences, 44307 Kaunas, Lithuania; linateee@gmail.com (L.B.); janina.petkeviciene@lsmuni.lt (J.P.); 2Department of Nutrition and Food Science, Campus of Melilla, University of Granada, 52001 Melilla, Spain; celiarp@ugr.es; 3Biomedical Research Centre, Institute of Nutrition and Food Technology (INYTA) ‘José Mataix’, University of Granada, 18071 Granada, Spain

**Keywords:** nutrition, physical activity, alcohol consumption, body weight, COVID-19, quarantine

## Abstract

The COVID-19 quarantine has caused significant changes in everyday life. This study aimed to evaluate the effect of the quarantine on dietary, physical activity and alcohol consumption habits of Lithuanians and the association between health behaviours and weight changes. An online cross-sectional survey was carried out among individuals older than 18 years in April 2020. The self-administered questionnaire included health behaviour and weight change data. Altogether 2447 subjects participated in the survey. Almost half of the respondents (49.4%) ate more than usual, 45.1% increased snacking, and 62.1% cooked at home more often. Intake of carbonated or sugary drinks, fast food and commercial pastries decreased, while consumption of homemade pastries and fried food increased. A decrease in physical activity was reported by 60.6% of respondents. Every third (31.5%) respondent, more often those already with overweight, gained weight. Multivariate logistic regression analysis showed that the higher odds of weight gain were associated with females, older age, increased consumption of sugary drinks, homemade pastries and fried food, eating more than usual, increased snacking, decreased physical activity and increased alcohol consumption. Our data highlighted the need for dietary and physical activity guidelines to prevent weight gain during the period of self-isolation, especially targeting those with overweight and obesity.

## 1. Introduction

The novel severe acute respiratory syndrome coronavirus 2 (SARS-CoV-2), which causes coronavirus disease (COVID-19), first appeared in December 2019 in Wuhan, China, and quickly spread worldwide. On 30 January 2020, the Director-General of the World Health Organization (WHO) declared the outbreak of COVID-19 to be a public health emergency of international concern and a set of recommendations was issued [1]. In Lithuania, the first case of COVID-19 was diagnosed on 28 February 2020. As the number of cases was increasing rapidly, the Lithuanian Government decided to declare quarantine from 16 to 30 March [2]. This was extended several times and ended on 16 June. All public indoor and outdoor gatherings were prohibited. Educational institutions began to work remotely. Shops excluding grocery shops and pharmacies were closed. Restaurants and bars were also closed, leaving the option for food takeaway. This situation forced people to cook at home. Social isolation disrupted daily routine. Increased sedentary behaviour and screen time, and limited food availability and choice could lead to changes in nutrition habits and weight gain [3]. Quarantine caused many economic, social and health problems. Some workers were forced to take unpaid vacation. They lost income and felt uncertain [4]. Stress is related to higher energy intake, ‘food craving’, unhealthy dietary patterns high in fat and sugar and higher alcohol consumption, which can lead to weight gain and increased risk of obesity [5,6]. Physical inactivity may also contribute to weight and body fat gain, causing various cardio-metabolic disorders [7]. Recent studies showed that obesity is associated with more severe COVID-19 illness and outcomes [8,9]. Existing evidence suggests that an unhealthy diet leads to chronic inflammation and impaired defence against viruses [10]. WHO experts developed specific recommendations on nutrition and physical activity for quarantine isolation [11]. However, in this unprecedented time with many economic, social, and psychological problems, it could be complicated to follow such recommendations. Our study aimed to investigate the effect of the COVID-19 quarantine on dietary, physical activity and alcohol consumption habits of Lithuanians and the association between health behaviour and weight changes.

## 2. Materials and Methods 

### 2.1. Study Sample and Data Collection

This study is a part of the international COVIDiet project led by researchers from the University of Granada, Spain [12]. The data collected in Lithuania were analysed. The online cross-sectional survey was carried out among individuals older than 18 years. Age was the only inclusion/exclusion criterion. The survey began on 14 April, a month after COVID-19 quarantine started in Lithuania, and lasted two weeks. A self-administered anonymous web-based questionnaire was developed to collect data. The link to the questionnaire was distributed using social media such as Facebook, web-pages of some institutions, which agreed to participate, social networking sites, and emails with the aim of reaching the greatest number of participants from all municipalities of Lithuania. In total, 2447 individuals (2149 females and 298 males) participated in the Lithuanian survey.

### 2.2. Questionnaire and Variables

The study questionnaire included 44 questions regarding various sociodemographic criteria, the Mediterranean Diet Adherence Screen (MEDAS) [12,13], and changes in eating behaviours during the quarantine: snacking and intake of some foods with possible answers: ‘Higher’, ‘Lower’ or ‘As usual’; perception of eating more during the quarantine with possible answers ‘Yes’ or ‘No’; changes in physical activity with possible answers: ‘It has increased’, ‘It has decreased’, ‘It remains as usual’, ‘I do not practice physical activity’; and changes in alcohol consumption with possible answers: ‘Yes, my intake of alcoholic beverages is higher’, ‘No, my intake of alcoholic beverages is lower’, ‘My intake of alcoholic beverages remains as usual’. The question about weight change was: ‘Have you gained weight during the quarantine’ with possible answers: ‘Yes’, ‘No’, ‘I don’t know’. By weight change, the respondents were grouped in those who gained weight (‘Yes’) and others (‘No’ or ‘I don’t know’). Self-reported current weight and height were used to calculate body mass index (BMI) as weight in kilograms divided by height in meters squared. According to WHO guidelines, overweight was defined as BMI 25–29 kg/m^2^ and obesity as BMI ≥ 30 kg/m^2^ [14]. 

The respondents were grouped into three age groups: 18–35 years, 36–50 years and 51 years and older. By education level, respondents were categorized into two groups: with university education and lower level of education. 

### 2.3. Data Analysis

Data analysis was performed using the statistical package IBM SPSS Statistics for Windows, Version 20.0 (IBM Corp.: Armonk, NY, USA, released 2011). The categorical variables were presented as percentages and compared using the chi-square test and z test with Bonferroni correction. Univariate and multivariate logistic regression analysis was used to evaluate the associations between weight gain during COVID-19 quarantine (dependent variable) and social factors, nutrition, physical activity and alcohol consumption habits.

### 2.4. Ethical Issues 

The international COVIDiet study was approved by the Research Ethics Committee of the University of Granada (1526/CEIH/2020) and was registered in ClinicalTrials.gov (https://clinicaltrials.gov/ct2/show/NCT04449731). Additionally, the Lithuanian study was approved by the Bioethics Centre of the Lithuanian University of Health Sciences. No personal data was collected and participation was voluntary and anonymous. The participants were informed about the objectives of the study.

## 3. Results

The main characteristics of the study population are presented in Table 1. The majority of the participants were females, 18–35 years old and having a university education. Prevalence of overweight was 27.8% and obesity 12.4%. Every third respondent (31.5%) gained weight during the quarantine.

A considerable number of participants reported that they changed their eating habits during the quarantine (Table 2). Almost half of the respondents (49.4%) answered that they ate more than usual. A similar proportion of participants (45.1%) increased snacking. However, 62.1% of respondents replied that they cooked at home more often than before quarantine. Every fifth participant (20.6%) increased consumption of fried food. Fast food became less popular, as 41.3% of respondents decreased its consumption. More than one-third of participants (37.7%) reported that they increased consumption of homemade pastries. Intake in carbonated and sugary drinks was decreased by 19.4% of respondents and consumption of commercial pastries by 26%. More people reduced read meat and fish consumption than increased, while a higher proportion of participants increased fruit and vegetable consumption than decreased. 

Most of the participants (69.9%) declared that their intake of alcoholic beverages (wine, beer, strong alcoholic drinks) remained the same as before the COVID-19 quarantine. The proportion of those who increased (14.2%) and decreased (15.9%) consumption of alcohol was similar. A large number of participants (60.6%) reported that they reduced physical activity (Table 2).

The changes in health behaviours differed according to changes in body weight (Table 3). Diet of those who gained weight changed in a more negative direction compared to those who reported no changes or did not notice the increase in body weight. More weigh-gaining respondents reduced fruit and vegetable consumption and increased red meat, carbonated or sugary drinks, pastries, fast food and fried food consumption. The majority of participants (84.3%) whose weight increased declared that they ate more compared to 33.4% of individuals without changes in body weight. More weigh-gaining people increased snacking and cooked more often than before quarantine. Increase in alcohol consumption was reported by every fifth participant (20.9%) who gained weight, and every tenth (11.2%), who did not. A large proportion of weight-gaining respondents (85.2%) decreased their physical activity during quarantine (Table 3). 

The weight gain was associated with BMI of participants. Those with higher BMI gained weight more often compared to those having normal BMI (Figure 1).

The binary univariate logistic regression showed that women were more likely to gain weight compared to men (Table 4). No associations were found with the age and education of respondents. The odds of weight gain were higher for those who decreased consumption of fresh fruits and vegetables and increased consumption of red meat, carbonated or sugary drinks, pastries, snacks, fast or fried food and ate more than usual. Increased alcohol consumption and decreased physical activity were also related to higher odds of weight gain. In multivariate logistic regression analyses, the association of weight gain with increased intake of carbonated or sugary drinks, homemade pastries and fast food, increased snacking, eating more than usual, decreased physical activity and increased alcohol consumption remained statistically significant. The odds of weight gain also increased with age. 

## 4. Discussion

Our study demonstrated that Lithuanians changed their eating habits during the quarantine. Almost half of the respondents reported that they ate more than usual, increased snacking and cooked more often at home. The intake of foods changed in positive and negative directions. More respondents increased than reduced fruit and vegetable consumption. The intake of carbonated and sugary drinks, fast food, and commercial pastries decreased; however, the consumption of homemade pastries and fried food increased. Negative changes in eating, physical activity, and alcohol consumption habits were associated with weight gain which was reported by one third of respondents.

Our data support previous findings showing that restrictions during COVID-19 increased the number of main meals, frequency of cooking, and snacking [12,15,16,17,18]. Spending more time on cooking offers the possibilities to eat healthier, try new recipes and enjoy food without rushing. To remind people of the most important principles of healthy nutrition, WHO experts developed recommendations on food and nutrition during self-quarantine [1]. However, restrictions during COVID-19 pandemic decreased the availability of some foods. People were urged to stay at home, all restaurants and bars were closed, and only some of them offered food takeaway. Ordering food online was overloaded [12]. The variety of food products in grocery stores has declined [12,19]. A Spanish study highlighted that people reported difficulties of finding meat in the supermarkets and grocery stores, which could be related to a decrease in consumption of meat during quarantine [12]. At the beginning of the quarantine, people experienced panic and uncertainty, which led to extra buying of all long-lasting food products and caused a shortage of grain and some other products in grocery stores [19]. 

Social isolation with or without loneliness and boredom can be associated with a negative impact on nutrition habits, obesity, physical activity, and poor physical and mental health [6,15,18,20,21]. Some authors showed that high-level of depression was related to poor diet, increased intake of saturated fat, energy-dense and salty foods [6,22]. Women who tended to act impulsively in a stressing environment were more likely to eat more candies [23]. People with anxiety consumed snacks 2.45 times more often [24]. The studies performed during the COVID-19 pandemic revealed more frequent and increased consumption of sweets, biscuits and cakes, decreased intake of fruits and vegetables and increased consumption of frozen and canned foods, more frequent cooking and eating out of control [15,16,17,18,24,25]. On the other hand, some positive changes such as decreased consumption of processed meat, carbonated or sugary drinks and increased consumption of fresh fruits and vegetables, fish, legumes and white meat were identified [12,16]. Our data support the previous findings that both positive and negative changes in dietary habits were observed during the quarantine. 

It is well documented that physical activity has many positive health effects [26]. Together with the changes in diet, a strong impact of quarantine on physical activity was observed. Many studies showed that all levels of physical activity decreased and sitting or screen time increased [12,15,16,17,18,22,24]. Few studies found opposite results, showing an increase of physical activity during COVID-19 crisis [16,27]. These differences may be related to different governmental policy on movement restrictions during this period. 

The data on alcohol consumption is controversial. Some studies reported an increase in alcohol consumption and higher drinking tendency among alcohol addicts [17,25], while other studies reported decrease in binge drinking and alcohol consumption [12,15,16,22]. A British study found decreased alcohol consumption only in the younger cohort [27]. Our study demonstrated similar proportions of those who increased and decreased alcohol consumption. 

Evidence suggests that even a small positive energy balance over time is sufficient to cause weight gain in many individuals [28]. Our data showed that those respondents who gained weight were more likely to change their diet in an unhealthy direction, decrease physical activity, and increase alcoholic beverages consumption than those who did not report changes in body weight. Furthermore, more respondents with higher BMI at the beginning of quarantine gained weight compare to those having a normal weight. 

Previous studies have also reported weight gain during the quarantine [16], providing estimated weight changes from 1.6 kg to 2.07 kg in different countries [16,17,18,24,25]. People following an unhealthy diet had 4.5 times higher odds of weight increase [24]. Weight gain was associated with higher consumption of most foods: meat, dairy, fast foods, and even fruits, vegetables and legumes [25]. An Italian study found no associations between BMI and healthy food intake, while higher consumption of junk food such as snacks, dressings, sweet beverages and sweets was associated with higher BMI [16]. Lower exercise, consumption of snacks, unhealthy foods, cereals, and sweets were associated with significant weight gain in adults with obesity a month after the beginning of the quarantine [18]. 

The latest studies report that patients with obesity are at increased risk of complications of COVID-19 infection and higher mortality rates [8,29]. As there is not yet effective treatment and vaccination against COVID-19 and there may be new virus outbreaks, healthy eating and physical activity promotion should be incorporated in government initiatives to reduce the negative effects of the COVID-19 pandemic [30]. Encouraging people to follow a balanced diet, especially targeting those with obesity, could help to maintain an effective immune system and may provide protection against infections and other diseases.

The strengths of our study include the online survey, which allowed us quickly to reach a sufficiently large sample of the population from different municipalities of the whole country during the quarantine. Additionally, we used the COVIDiet questionnaire prepared by researchers from the University of Granada for the international study so we will be able to compare our data with other countries in the future. Several limitations of our study should be mentioned for the accurate interpretation of its results. First, we used non-random sampling to reach participants. The majority of respondents were women and people with higher education. The higher participation of women was also reported by other researchers [12,18]. However, our study covered all municipalities of Lithuania and a large age interval. Furthermore, we used self-reported online survey data which may be less reliable and biased. BMI was calculated from self-reported data on weight and height, which could lead to underestimation of overweight and obesity. A question about weight loss was not included in the questionnaire. Finally, we did not collect data on employment status and stress during the quarantine. Stress could impact eating and alcohol consumption habits. 

## 5. Conclusions

COVID-19 quarantine caused both positive and negative changes in nutrition, physical activity, and alcohol consumption habits in the Lithuanian population. Weight gain was associated with unhealthy dietary changes, decreased physical activity, and increased alcohol consumption. Our data highlighted the need for investigations into which dietary and physical activity guidelines might be appropriate to help counteract potentially negative impacts of COVID-19 on health behaviour and body weight.

## Figures and Tables

**Figure 1 nutrients-12-03119-f001:**
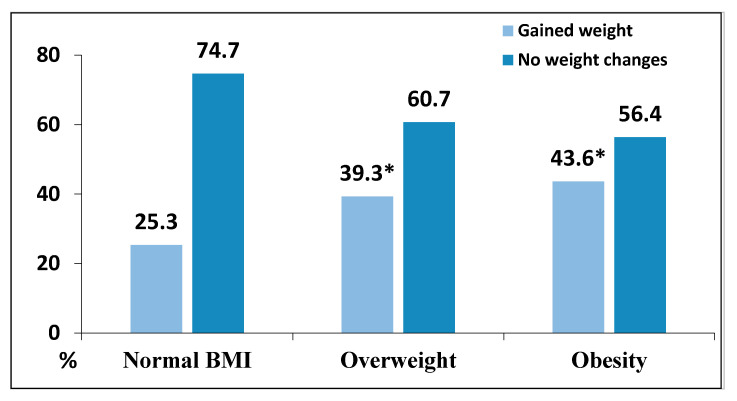
Association between body mass index (BMI) of the participants and weight gain during the quarantine (%). * *p* < 0.05 compared with normal BMI (z test with Bonferroni correction).

**Table 1 nutrients-12-03119-t001:** Characteristics of the study population (%).

Characteristics	n	%
**Sex**		
Male	298	12.2
Female	2149	87.8
**Age groups**		
18–35	982	40.1
36–50	898	36.7
≥51	567	23.2
**Education**		
University	1738	71.0
Lower	709	29.0
**Body mass index**		
<25 kg/m^2^	1458	59.8
25–29 kg/m^2^	679	27.8
≥30 kg/m^2^	303	12.4
**Weight changes during the quarantine**		
Gained	771	31.5
No changes/didn’t know	1676	68.5

**Table 2 nutrients-12-03119-t002:** Changes in health behaviours during the quarantine (%).

Health Behaviours	Changes		
	Increased	Decreased	Remains as Usual
**Food intake**			
Vegetables	18.8	15.0	66.2
Fruits	22.1	14.7	63.2
Pulses	9.1	8.5	82.4
Fish-seafood	7.5	14.3	78.3
Red meats, hamburgers, sausages	12.2	17.9	69.9
Carbonated and/or sugary beverages (soda, cola, tonic, bitter)	8.5	19.4	72.0
Commercial (non-homemade) pastries such as cookies, custards, sweets	18.9	26.0	55.2
Homemade pastries such as cookies, custards, sweets or cakes	37.7	11.5	50.8
Fast food	6.7	41.3	51.9
Fried food	20.6	8.3	71.1
**Eating habits**			
Snacking	45.1	9.8	45.1
Cooking more often than before the quarantine	62.1 (yes)	1.3 (no, less often)	36.5 (as usual)
Eat more than usual	49.4 (yes)	-	50.6 (as usual)
**Other lifestyle habits**			
Alcoholic beverages consumption	14.2	15.9	69.9
Physical activity	14.3	60.6	19.3

**Table 3 nutrients-12-03119-t003:** The proportion of participants who reported changes in health behaviours by changes in weight (%).

Changes in Health Behaviours	Weight Change				*p*-Value *
	Gained		No Changes/Doesn’t Know		
	*N*	%	*N*	%	
**Food intake**					
Vegetables decreased	163	21.1	203	12.1	0.001
Fruits decreased	144	18.7	216	12.9	0.001
Red meats, hamburgers, sausages increased	157	20.4	141	8.4	0.001
Carbonated and/or sugary beverages increased	116	15.0	93	5.5	0.001
Commercial pastries increased	241	31.3	221	13.2	0.001
Homemade pastries increased	405	52.5	517	30.8	0.001
Fast food increased	95	12.3	70	4.2	0.001
Fried food increased	251	32.6	253	15.1	0.001
**Eating habits**					
Snacking increased	566	73.4	537	32.0	0.001
Cooking more often than before the quarantine	548	71.1	972	58.0	0.001
Eat more than usual	650	84.3	560	33.4	0.001
**Other lifestyle habits**					
Alcoholic beverages consumption increased	161	20.9	187	11.2	0.001
Physical activity decreased	657	85.2	967	57.7	0.001

* z test with Bonferroni correction was used for multiple comparisons.

**Table 4 nutrients-12-03119-t004:** Odds Ratios (OR) for the likelihood of weight gain by socio-demographic variables and changes in health behaviours during the quarantine.

Variables	Univariate Analysis		Multivariate Analysis	
	OR (95% CI)	*p*-Value	OR (95% CI)	*p*-Value
**Sex**				
Male	1		1	
Female	1.37 (1.04–1.80)	0.025	1.52 (1.08–2.14)	0.015
**Age groups**				
18–35	1		1	
36–50	1.14 (0.94–1.39)	0.173	1.36 (1.06–1.73)	0.014
≥51	1.08 (0.86–1.35)	0.483	1.80 (1.35–2.39)	0.001
**Education**				
University	1		1	
Lower	1.18 (0.93–11.42)	1.172	1.20 (0.96–1.51)	0.113
Intake of vegetables decreased *	1.94 (1.55–2.44)	0.001	1.12 (0.82–1.53)	0.484
Intake of fruits decreased *	1.55 (1.23–1.95)	0.001	1.12 (0.81–1.52)	0.530
Intake of red meat increased *	2.78 (2.17–3.56)	0.001	1.03 (0.75–1.40)	0.871
Intake of carbonated or sugary drinks increased *	3.01 (2.26–4.02)	0.001	1.44 (1.01–2.06)	0.049
Intake of commercial pastries increased *	2.99 (2.43–3.68)	0.001	1.20 (0.92–1.56)	0.184
Intake of homemade pastries increased *	2.48 (2.08–2.95)	0.001	1.56 (1.25–1.95)	0.001
Intake of fast food increased *	3.22 (2.33–4.44)	0.001	1.62 (1.09–2.43)	0.018
Intake of fried food increased *	2.71 (2.22–3.32)	0.001	1.15 (0.89–1.50)	0.287
Snacking increased *	5.85 (4.84–7.08)	0.001	1.55 (1.20–2.01)	0.001
Eating more than usual *	10.70 (8.60–13.23)	0.001	5.68 (4.30–7.51)	0.001
Physical activity decreased *	4.22 (3.38–5.27)	0.001	3.24 (2.53–4.16)	0.001
Alcohol consumption increased *	2.10 (1.66–2.64)	0.001	1.47 (1.11–1.95)	0.008

* Reference variable – no change in respective health behaviour or change in the opposite direction; OR—odds ratio; CI—confidence interval.

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
