# Peer review of "Associations between Changes in Health Behaviours and Body Weight during the COVID-19 Quarantine in Lithuania: The Lithuanian COVIDiet Study"

_nutrients, 2020, doi:10.3390/nu12103119_

Round 1

Reviewer 1 Report

Please go through and correct all instances of passive voice, grammatical errors, and formatting errors. In some cases, there are issues with omitted words like "have", "has", and "the". Please correct these omissions. You also occasionally mix up decimals and commas (e.g., line 99). Pick one style and make sure it's the one the journal takes. There are comma splices and unnecessary commas aplenty in your piece. 

Source all empirical claims. For example, on line 44, the sentence "Most of the Lithuanian people understood the seriousness of the problem" is left without a source. There are many cases of this throughout the paper and they must be removed or given a fitting source. I would not encourage you to go looking for fitting sources; you should instead be comfortable removing extraneous sentences and sentences including empirical claims left without evidence.

In line 71, there is an error. You state that "2149 individuals participated in the Lithuanian survey" and yet you list "2447 females and 298 males" as the sample sizes for each sex. By my count, this is a sample size of 2745. I think you erred by mixing up the female and total sample sizes, as indicated in your Table 1. 

In line 73, you use a colon to begin listing the question topics covered by the study but this needs to be rewritten. For instance, it could more clearly read "The study questionnaire included 44 questions regarding various sociodemographic criteria, the Mediterranean Diet Adherence Screen (MEDAS), and changes in eating behaviours during quarantine." This is probably not sufficient, but you could still do to reword it so the sentence doesn't require readers to fill in the gaps you neglected to state. 

Since you are using SPSS, please provide your analysis syntax for reproducibility reasons. There is no reason not to include this. 

In line 94, you state that you used the - typical - p = 0.05 threshold. Please do not. If you would like your results to be interpreted as a 5% false-positive rate as opposed to the much higher rate your data will yield with p = 0.05 (see, e.g., Colquhoun, 2014), you need to scale it. Here is the basic scaling function from Naaman (2016) as a snippet of R code: 

NP <- function(N, S = 2) { NP = 1-pnorm(qnorm(1-(N^(-6/5))/S)) return(NP)}

The NP function takes your sample size, N, and the number of sides for the p value, S (default = 2, as is typical) and returns an appropriate p value to maintain a 5% false positive rate. There are small sample and Bonferroni-corrected versions of this (and there's more you can do for unbalanced groups), but it is unnecessary to invoke those as this is likely strict enough and your sample is not particularly small. For n = 2447, your critical p value should be 0.00004291501. If you wish to maintain the interpretation of a p value of 0.05 (i.e., >5% false positive rate), you can scale like Good (1988) advised. Assuming you want a p value comparable to 0.05 at n = 100, your p value for n = 2447 would be 0.05/sqrt(2447/100) = 0.01. Whatever you choose, justify your alpha and do so without saying that you merely used what was typical (i.e., 0.05).

The grouping for weight change is extremely dubious. It only has the categories "Gained" and "No + I don't know"; are you really saying that no one lost weight in the sample? If so, please clarify in the paper. That is an interesting tidbit. If it was not assessed, that is a serious oversight that needs to be addressed in later iterations of this study.

You report that there were significant changes in food intake during quarantine and that those results are located in Table 2, but Table 2 contains no p values. Please add these or drop any reference to the significance of the changes and qualify your statements. If all the values are below your significance threshold, just specify that. If only a few are not, you could just specify that too. Whatever your eventual choice, make sure to show the things you wrote you found. 

When stating, for example, that an increase in alcohol consumption was reported by every fifth participant who gained weight or every tenth who did not, specify the percentages, because they are unlikely to be 20 and 10% (Table 3 shows they are not), even though they are close enough to verbally go by "every fifth" and "every tenth". Readers often skim tables, so this is important. 

You perform many significance tests but do not correct them for multiple comparisons. Do this appropriately. Remember that even if your results are not significant at some threshold, you may still state that they, cumulatively, pointed in the same direction and that future studies should assess whether the results replicate. It is still worth publishing even if results are not significant. 

It is not clear what the reference variable is for the intake variables in Table 4. Seeing your model syntax would be very informative here in order to see if your model is properly specified. If possible, make an effort to provide your data, either as a supplement or on osf.io or a comparable data sharing website. Since there is not personal information attached to your survey data, sharing it should be imminently feasible.

Your conclusion must be rewritten. Your analyses were exploratory. They do not highlight the need for any sort of dietary or exercise guidelines. On the other hand, they can be used to support the need for additional investigations into those subjects. After, say, a successful intervention on those fronts, you will have highlighted the need for them. Otherwise, your conclusions are overstated in a manner that - if it ends up being advice to policymakers - could be costly without good reason. This may seem like a pedantic point, but your conclusions about highlighted needs are too open-ended. Were Lithuania to follow the example of the United States or the United Kingdom, for example, they could produce harmful guidelines. To highlight a need for guidelines is to suggest the need for studies into what types of guidelines work, which is a different topic entirely. To reiterate this in different words, reduce the potency of the call-to-action in your conclusion since it is not justified by this piece of research, even if this might help to motivate the appropriate type of research for those statements. A better statement would be as follows: "Our data highlighted the need for investigations into what dietary and exercise guidelines might be appropriate to help offset potentially negative impacts of COVID on these determinants of wellbeing."

Overall, my recommendation is a round of modestly extensive revisions followed . I am only calling for major revisions so that the manuscript is sent back to me so that I might check it over after the revisions I asked for are made. I want this work to be as strong as possible and I want the conclusions to be warranted by the data and analytic choices of the authors, hence my comments. Cheers. 

https://royalsocietypublishing.org/doi/full/10.1098/rsos.140216

https://projecteuclid.org/euclid.ejs/1464710240

https://projecteuclid.org/euclid.ss/1177012754

Author Response

Reviewer  comments

We would like to thank you for your efforts and time to read and revise our manuscript. We appreciate your comments and suggestions. We hope that we have successfully addressed all of the concerns raised, and we believe that the manuscript has been substantially improved. Our detailed responses to the comments and the description of the changes we have made to the manuscript are provided below.

Point 1: Please go through and correct all instances of passive voice, grammatical errors, and formatting errors. In some cases, there are issues with omitted words like "have", "has", and "the". Please correct these omissions.

Response 1: Thank you for the comment. We do acknowledge our imperfections in the English language, although we put a lot of effort into this question. A colleague whose native language is English edited the manuscript. Grammatical and style errors were corrected.

 Point 2: You also occasionally mix up decimals and commas (e.g., line 99). Pick one style and make sure it's the one the journal takes.

Response 2: Thanks for noticing an error. It was corrected.

Point 3: Source all empirical claims. For example, on line 44, the sentence "Most of the Lithuanian people understood the seriousness of the problem" is left without a source. There are many cases of this throughout the paper and they must be removed or given a fitting source. I would not encourage you to go looking for fitting sources; you should instead be comfortable removing extraneous sentences and sentences including empirical claims left without evidence.

Response 3: As reviewer suggested, we removed some sentences including empirical claims (lines 43; 199-200)

Point 4: In line 71, there is an error. You state that "2149 individuals participated in the Lithuanian survey" and yet you list "2447 females and 298 males" as the sample sizes for each sex. I think you erred by mixing up the female and total sample sizes, as indicated in your Table 1.

Response 4: We are sorry for this typing error. It was corrected: ‘In total, 2149 individuals (2447 females and 298 males) participated in the Lithuanian survey’ (lines 65-66)

Point 5: In line 73, you use a colon to begin listing the question topics covered by the study but this needs to be rewritten. For instance, it could more clearly read "The study questionnaire included 44 questions regarding various sociodemographic criteria, the Mediterranean Diet Adherence Screen (MEDAS), and changes in eating behaviours during quarantine." This is probably not sufficient, but you could still do to reword it so the sentence doesn't require readers to fill in the gaps you neglected to state.

Response 5: Thank you for your suggestion. We corrected the sentence accordingly. (lines 68-70)

Point 6: Since you are using SPSS, please provide your analysis syntax for reproducibility reasons. There is no reason not to include this.

Response 6: We performed the statistical analysis using the SPSS user interface and choosing the only standard commands from the menu. No scripts were used nor created for calculations, so there is, unfortunately, nothing that we could include. In SPSS, all calculations could be easily repeated without additional information.

Point 7: In line 94, you state that you used the - typical - p = 0.05 threshold. Please do not. If you would like your results to be interpreted as a 5% false-positive rate as opposed to the much higher rate your data will yield with p = 0.05 (see, e.g., Colquhoun, 2014), you need to scale it. Here is the basic scaling function from Naaman (2016) as a snippet of R code:

NP <- function(N, S = 2) { NP = 1-pnorm(qnorm(1-(N^(-6/5))/S)) return(NP)}

The NP function takes your sample size, N, and the number of sides for the p value, S (default = 2, as is typical) and returns an appropriate p value to maintain a 5% false positive rate. There are small sample and Bonferroni-corrected versions of this (and there's more you can do for unbalanced groups), but it is unnecessary to invoke those as this is likely strict enough and your sample is not particularly small. For n = 2447, your critical p value should be 0.00004291501. If you wish to maintain the interpretation of a p value of 0.05 (i.e., >5% false positive rate), you can scale like Good (1988) advised. Assuming you want a p value comparable to 0.05 at n = 100, your p value for n = 2447 would be 0.05/sqrt(2447/100) = 0.01. Whatever you choose, justify your alpha and do so without saying that you merely used what was typical (i.e., 0.05).

Response 7: We thank the reviewer for the interesting references on the adjustment of p-value. They will be very useful to us in the future. However, the limited time given for the manuscript corrections was not enough to properly educate ourselves in the subject. Being a part of the international survey also does not allow as to use different statistical analysis of the data. So, we decided to keep the widely used statistical methodology, but we removed some claims about significance. (lines 90-91)

Point 8: The grouping for weight change is extremely dubious. It only has the categories "Gained" and "No + I don't know"; are you really saying that no one lost weight in the sample? If so, please clarify in the paper. That is an interesting tidbit. If it was not assessed, that is a serious oversight that needs to be addressed in later iterations of this study.

Response 8: In the questionnaire prepared by the researchers from the University of Granada for international study, the question about weight change was: ‘Have you gained weight during the quarantine’. It had only three possible answers: ‘Yes’, ‘No’ and ‘I do not know’. The answer or a separate question about weight loss was not included in the questionnaire. Now we added the question about weight change to the methods section (lines 75-76). Following the reviewer’s suggestion, we mentioned the absence of a question on weight reduction as a limitation in the discussion section: ‘The question about weight loss was not included in the questionnaire.’ (lines 233-234).

We will take this remark into account when planning future research.

 Point 9: You report that there were significant changes in food intake during quarantine and that those results are located in Table 2, but Table 2 contains no p values. Please add these or drop any reference to the significance of the changes and qualify your statements. If all the values are below your significance threshold, just specify that. If only a few are not, you could just specify that too. Whatever your eventual choice, make sure to show the things you wrote you found.

Response 9: In Table 2, we present only the distribution according to the answers to the questions. It is the subjective opinion of the respondents about what they have changed. Unfortunately, there is nothing to compare and there is no need to calculate p values. To avoid confusion, we changed wording in the sentence: ‘A considerable number of participants reported that they changed their eating habits during the quarantine (Table 2). ‘ (line 106)

Point 10: When stating, for example, that an increase in alcohol consumption was reported by every fifth participant who gained weight or every tenth who did not, specify the percentages, because they are unlikely to be 20 and 10% (Table 3 shows they are not), even though they are close enough to verbally go by "every fifth" and "every tenth". Readers often skim tables, so this is important.

Response 10: Following the reviewer’s suggestion, we added the percentage (lines 110 and 129)

Point 11: You perform many significance tests but do not correct them for multiple comparisons. Do this appropriately. Remember that even if your results are not significant at some threshold, you may still state that they, cumulatively, pointed in the same direction and that future studies should assess whether the results replicate. It is still worth publishing even if results are not significant.

Response 11: We needed to adjust for multiple comparisons only in calculations for Fig.1 where the proportion of respondents with weight gain were presented according to BMI (normal weight, overweight, and obesity) and for Table 3. All calculations were adjusted for multiple comparisons using z test with Bonferroni corrections. This is a standard command from the menu in SPSS. We mentioned it in the footnote of Fig.1. Our mistake is not to mark the adjustment in the footnote of Table 3. We corrected this mistake: ‘*z test with Bonferroni corrections was used for multiple comparisons.’ (line 134)

No p-value calculations were performed for Table 1 and Table 2 because only the distribution according to the answers to the questions without any comparison are presented in these tables.

Point 12: It is not clear what the reference variable is for the intake variables in Table 4. Seeing your model syntax would be very informative here in order to see if your model is properly specified. If possible, make an effort to provide your data, either as a supplement or on osf.io or a comparable data sharing website. Since there is not personal information attached to your survey data, sharing it should be imminently feasible.

Responce 12: The reference variable for change in health behaviour was ‘no change in respective health behaviour or change in the opposite direction’. We included this explanation in the footnote of Table 4 (line 153).

Logistic regression was performed using the only standard command from the menu in SPSS. The dependent dichotomised variable was ‘gained weight’ (1) and ‘did not gained weight or did know’ (0) as described in the methods section. Covariates were the variables included in Table 4. No additional syntax was used. We think the explanation what the reference variable for change in health behaviour is will help to understand the information provided in table 4. Knowing this information, it is easy to repeat the logistic regression analysis in SPSS.

We are not able to provide all data because they are part of the international study. According to the agreement, we cannot provide data to third parties until the articles on common data will be published.

Point 13: Your conclusion must be rewritten. Your analyses were exploratory. They do not highlight the need for any sort of dietary or exercise guidelines. On the other hand, they can be used to support the need for additional investigations into those subjects. After, say, a successful intervention on those fronts, you will have highlighted the need for them. Otherwise, your conclusions are overstated in a manner that - if it ends up being advice to policymakers - could be costly without good reason. This may seem like a pedantic point, but your conclusions about highlighted needs are too open-ended. Were Lithuania to follow the example of the United States or the United Kingdom, for example, they could produce harmful guidelines. To highlight a need for guidelines is to suggest the need for studies into what types of guidelines work, which is a different topic entirely. To reiterate this in different words, reduce the potency of the call-to-action in your conclusion since it is not justified by this piece of research, even if this might help to motivate the appropriate type of research for those statements. A better statement would be as follows: "Our data highlighted the need for investigations into what dietary and exercise guidelines might be appropriate to help offset potentially negative impacts of COVID on these determinants of wellbeing."

Responce 13: We agree with your point of view and have changed the conclusions in line with your recommendations: ‘Our data highlighted the need for investigations into what dietary and physical activity guidelines might be appropriate to help counteract potentially negative impacts of COVID on health behaviours and body weight’.

Sincerely,

Vilma Kriaucioniene

Lithuanian University of Health Sciences, Medical Academy,

Faculty of Public Health, Lithuania

Reviewer 2 Report

Dear authors,

The study is well written, easy to read and to understand. I have some minor questions to address. Kind regards

  • Please divide the methods section in more subsections to clarify it (study design, sample, variables, procedures and data collection, data analysis, ethical issues).
  • Did you use any inclusion/exclusion criteria (not just age)? Please specify it.
  • Was it possible to answer "Loose weight?"? I think it is estrange that no one has loose weight as many people increased sport and healthy food in some countries.

Author Response

Thank you very much for reviewing our manuscript. We appreciate your comments and suggestions. We hope that we have successfully addressed all of the concerns raised, and we believe that the manuscript has been substantially improved. Our detailed responses to the comments and the description of the changes we have made to the manuscript are provided below.

Point 1: Please divide the methods section in more subsections to clarify it (study design, sample, variables, procedures and data collection, data analysis, ethical issues).

Response 1: As the reviewer suggested, we divided the methods section into the following subsections: 2.1. Study Sample and Data Collection; 2.2. Questionnaire and Variables; 2.3. Data Analysis and 2.4 Ethical issues

Point 2: Did you use any inclusion/exclusion criteria (not just age)? Please specify it.

Response 2: Age was the only inclusion/exclusion criterion. We added this sentence to the methods section (lines 60-61). On the other hand, it was online survey; thus, respondents were required to have a computer and internet connection.  

Point 3: Was it possible to answer "Loose weight?"? I think it is strange that no one has loose weight as many people increased sport and healthy food in some countries.

Response 3: In the questionnaire prepared by the researchers from the University of Granada for international study, the question about weight change was: ‘Have you gained weight during the quarantine’. It had only three possible answers: ‘Yes’, ‘No’ and ‘I do not know’. The answer or a separate question about weight loss was not included in the questionnaire. Now we added the question about weight change to the methods section (lines 75-76). Following the reviewer’s suggestion, we mentioned the absence of a question on weight reduction as a limitation in the discussion section: ‘The question about weight loss was not included in the questionnaire.’ (lines 233-234)

Sincerely,

Vilma Kriaucioniene

Lithuanian University of Health Sciences, Medical Academy,

Faculty of Public Health, Lithuania

Reviewer 3 Report

This manuscript focuses an important issue in our current daily life. I think one of the main advantages of this manuscript is the number of participants and the second one is the picture of eating behaviour during quarantine.

 I have some suggestions and answers that I would like to be answered as minor revisions:

First of all, please clarify the total number of participants. This information is not clear throughout the paper. In line 71, “In total, 2149 individuals (2447 females and 298 males) …”. The number of females is higher than the total sample.

Abstract

  • Lacks a brief introduction;
  • Clarify the sample size and the number of females and males.

Introduction

  • Clarify the sample size and the number of females and males and please change/correct the numbers.

Table 1

  • The total sample is 2447 for variables sex, age groups, education and weight changes (what is the right number?) and 2440 for BMI.

Results

  • Line 114: “The proportion … were …” - change to “The proportion … was …”;
  • Add space between Table 2 and the paragraph and Table 3 and the paragraph;
  • Line 137: “… were higher for those, …” - remove comma “… were higher for those …”.

Discussion

  • Lines 191-192: Please rewrite this sentence;
  • Lines 204-206: Please rewrite this sentence. Personally, I don´t like “A study of people with obesity …”;
  • Lines 217-218: How did you make this comparison?

Author Response

We would like to thank to you for reviewing our manuscript. We hope that we have successfully addressed all of the concerns raised. Our detailed responses to the comments and the description of changes we have made to the manuscript are provided below.

Point 1: First of all, please clarify the total number of participants. This information is not clear throughout the paper. In line 71, “In total, 2149 individuals (2447 females and 298 males) …”. The number of females is higher than the total sample.

Response 1: Thanks for noticing an error. It was corrected: ‘In total, 2447 individuals (2149 females and 298 males) participated in the Lithuanian survey.’ (lines 65-66)

Point 2: Abstract. Lacks a brief introduction.

Clarify the sample size and the number of females and males

Response 2: The abstract must not exceed 200 words. So, we were able to include only one sentence: ‘The COVID-19 quarantine caused significant changes in everyday life.’ (line 13)

The sample size was corrected: ‘Altogether 2447 subjects participated in the survey’ (line 17).

Point 3: Clarify the sample size and the number of females and males and please change/correct the numbers.

Response 3: It was corrected: In total, 2447 individuals (2149 females and 298 males) participated in the Lithuanian survey. (lines 65-66)

Point 4: Table 1. The total sample is 2447 for variables sex, age groups, education and weight changes (what is the right number?) and 2440 for BMI.

Response 4: The total sample is 2447 respondents. However, some people did not provided information about their weight. As they answered the questions about changes in health behaviours and body weight, they were not excluded from the analyses, except from the BMI calculation.

Point 5: Line 114: “The proportion … were …” - change to “The proportion … was”

Response 5: Thanks for noticing an error. It was corrected.

Point 6: Add space between Table 2 and the paragraph and Table 3 and the paragraph;

Response 6: We added the recommended spaces.

Point 7: Line 137: “… were higher for those, …” - remove comma “… were higher for those …”

Response 7: Comma was removed.

Point 7: Lines 191-192: Please rewrite this sentence.

Response 7: The sentence ‘Even not long-lasting changes of environmental and behavioural factors, leading to sedentary behaviours and adoption of unhealthy eating habits may cause weight gain’ was removed because it repeated the information given in the next sentence. (line 199-200).

Point 8: Please rewrite this sentence. Personally, I don´t like “A study of people with obesity …”

Response 8: The sentence ‘A study of people with obesity also found that unhealthy diet, such as excessive consumption of snacks, cereals and sweets, led to significant weight gain in 1 month after the beginning of the quarantine’ was changed: ‘Lower exercise, consumption of snacks, unhealthy foods, cereals, and sweets were associated with significant weight gain in adults with obesity a month after the beginning of the quarantine’ (line 213-215)

Point 9: Lines 217-218: How did you make this comparison?

Response 9: Thanks for the remark. In this article, we did not compare the data with our colleague’s data. We hope to do this in the future.

We changed the sentence: ‘Additionally, we used COVIDiet questionnaire prepared by the researchers from the University of Granada for the international study so we will be able to compare our data with other countries in the future’. (line 225-227).

Sincerely,

Vilma Kriaucioniene

Lithuanian University of Health Sciences, Medical Academy,

Faculty of Public Health, Lithuania
